# Computational and In Vitro Investigation of (-)-Epicatechin and Proanthocyanidin B2 as Inhibitors of Human Matrix Metalloproteinase 1

**DOI:** 10.3390/biom10101379

**Published:** 2020-09-28

**Authors:** Kyung Eun Lee, Shiv Bharadwaj, Umesh Yadava, Sang Gu Kang

**Affiliations:** 1Department of Biotechnology, Institute of Biotechnology, College of Life and Applied Sciences, Yeungnam University, 280 Daehak-Ro, Gyeongsan, Gyeongbuk 38541, Korea; keun126@ynu.ac.kr (K.E.L.); shiv@ynu.ac.kr (S.B.); 2Department of Physics, Deen Dayal Upadhyay Gorakhpur University, Gorakhpur, Uttar Pradesh 273009, India; u_yadava@yahoo.com; 3Stemforce, 313 Institute of Industrial Technology, Yeungnam University, 280 Daehak-Ro, Gyeongsan, Gyeongbuk 38541, Korea

**Keywords:** matrix metalloproteinases 1, epicatechin, proanthocyanidin B2, molecular dynamics simulation, zymography

## Abstract

Matrix metalloproteinases 1 (MMP-1) energetically triggers the enzymatic proteolysis of extracellular matrix collagenase (ECM), resulting in progressive skin aging. Natural flavonoids are well known for their antioxidant properties and have been evaluated for inhibition of matrix metalloproteins in human. Recently, (-)-epicatechin and proanthocyanidin B2 were reported as essential flavanols from various natural reservoirs as potential anti-inflammatory and free radical scavengers. However, their molecular interactions and inhibitory potential against MMP-1 are not yet well studied. In this study, sequential absorption, distribution, metabolism, and excretion (ADME) profiling, quantum mechanics calculations, and molecular docking simulations by extra precision Glide protocol predicted the drug-likeness of (-)-epicatechin (−7.862 kcal/mol) and proanthocyanidin B2 (−8.145 kcal/mol) with the least reactivity and substantial binding affinity in the catalytic pocket of human MMP-1 by comparison to reference bioactive compound epigallocatechin gallate (−6.488 kcal/mol). These flavanols in docked complexes with MMP-1 were further studied by 500 ns molecular dynamics simulations that revealed substantial stability and intermolecular interactions, viz. hydrogen and ionic interactions, with essential residues, i.e., His218, Glu219, His222, and His228, in the active pocket of MMP-1. In addition, binding free energy calculations using the Molecular Mechanics Generalized Born Surface Area (MM/GBSA) method suggested the significant role of Coulomb interactions and van der Waals forces in the stability of respective docked MMP-1-flavonol complexes by comparison to MMP-1-epigallocatechin gallate; these observations were further supported by MMP-1 inhibition assay using zymography. Altogether with computational and MMP-1–zymography results, our findings support (-)-epicatechin as a comparatively strong inhibitor of human MMP-1 with considerable drug-likeness against proanthocyanidin B2 in reference to epigallocatechin gallate.

## 1. Introduction

Skin, the outermost protective covering of the human body, is composed of a multilayered structure, i.e., oversimplified as an underlying matrix composed of a supporting dermis and a functional epithelium at the outer surface. The epidermis region of the skin is enriched with keratinocytes that differentiate and stratify towards the outer skin surface to act as a skin barrier [1]. Also, the skin dermis is abundant with fibroblast cells that secrete structural components to support the extracellular matrix [2]. Fibroblasts are well documented for their significant reparative effect in the lower layer of skin connective tissue to regulate extracellular matrices, interstitial fluid volume, and wound healing [3,4]. The qualitative and quantitative cellular alternations mediated by fibroblasts in the dermal extracellular matrix (ECM) are recorded as the most prominent characteristic of elderly skin senescence [5]. For instance, aging fibroblasts were reported to cause rapid synthesis and secretion of matrix metalloproteinases 1 and 3 (MMP-1 and MMP-3) to advance the skin aging process via degradation of the skin’s collagen matrix [6]. These MMPs, which belongs to the family of Ca^2+^-containing and Zn^2+^-dependent endopeptidases, are actively secreted by the keratinocytes and the dermal fibroblasts under the influence of multiple stimuli such as oxidative stress, Ultraviolet (UV) radiation, and cytokines [7,8]. To date, at least 28 different types of MMPs have been discovered that play an important role in various pathophysiological processes, including photoaging, wound healing, skeletal growth and remodeling, arthritis, inflammation, angiogenesis, and cancer [9,10,11]. Enzymes of the matrix metalloproteinase (MMP) family, especially MMP-1, play an important role in skin aging [12]. Previous studies reported that oxidative stress produced in the epidermal cells by UV treatment activates the mitogen-activated protein kinase (MAPK) signaling pathway and the transcription factor activator protein 1 (AP-1) to promote skin aging by enhanced expression of MMP-1 into the ECM [13,14]. The proteolytic activity of MMP-1 enzyme resides in the catalytic domain but it requires the hemopexin domain to cleave the collagen [15]. Hence, several studies have suggested effective approaches to stimulate epidermal cell growth along with efficient remedies to suppress aging fibroblasts synthesizing and secreting matrix metalloproteinases (MMPs) [16]. 

Recently, bioactive molecules from plants have been widely used as cosmeceutical ingredients because of their prime property to slow down the rate of intrinsic skin aging processes and to diverge the extrinsic ones [17]. Many bioactive compounds have been discovered that can suppress the UV-induced expression of MMPs. For example, the most successful polyphenolic inhibitor against MMPs activity is epigallocatechin gallate (EGCG), which showed IC_50_ values close to 10 µM for MMP-2 and MMP-9 [18], while an inhibition constant (Ki) at 9.56 ± 1.67 µM was reported for MMP-1 in human macrophages [19]. Furthermore, proanthocyanidins, also known as condensed tannins, are a class of polymeric phenolic compounds that mainly consist of polyhydroxy flavan-3-ol units, i.e., catechin, (-)-epicatechin, gallocatechin, and epigallocatechin units; these units are mainly linked by the C4→C8 bond, but the C4→C6 linkage also exists in some natural compounds [20,21]. Among the distinct types of proanthocyanidins, the polyphenolic compounds exclusively consisting of (-)-epicatechin units are abundant in plants also designated as procyanidins. Most of the plant-based foods like fruits, berries, nuts, beans, cereals, spices, and beverages (wine and beers) were documented to contain exclusively the homogenous B-type procyanidins [22]. For example, proanthocyanidin B2, a dimer structure of (-)-epicatechin, i.e., (-)-epicatechin-(4β→8)–(-)-epicatechin structure, is known for a broad spectrum of pharmacological and medicinal properties including anti-apoptosis, anti-inflammatory, and antiatherosclerosis [23,24,25]. However, to the best of our knowledge, (-)-epicatechin and proanthocyanidin B2 are not yet studied via molecular simulations and tested in vitro for their inhibition activity against human MMP-1. Thus, this study was designed to understand the mechanistic inhibition of human MMP-1 by (-)-epicatechin and proanthocyanidin B2 against positive inhibitor EGCG using computational approaches and were examined in vitro via the zymography method for the inhibition of the human MMP-1 secreted by fibroblasts.

## 2. Methodology

### 2.1. Receptor and Ligands Collection

Three-dimensional (3D) structure of the human fibroblast collagenase (MMP-1, PDBID:1HFC) [26] as a receptor was retrieved from RCSB protein database (http://www.rcsb.org) [27] which contained 19 kDa active fragments of the human fibroblast collagenase refined at 1.56 Å resolution with a R-factor of 17.4%, included methylamino-phenylalanyl-leucyl-hydroxamic acid inhibitor, 88 water molecules, and three metal atoms (two zincs and a calcium) [26]. Also, 3D structures of the selected flavanols, i.e., (-)-epicatechin (CID:72276), proanthocyanidin B2 (CID:122738), and reference compound epigallocatechin gallate (EGCG; CID:65064), were retrieved from PubChem database (https://pubchem.ncbi.nlm.nih.gov) [28].

### 2.2. ADME and Quantum Chemical Calculations

Initially, the flavanols (-)-epicatechin, proanthocyanidin B2, and EGCG were studied for their physiochemical properties using the SwissADME server (http://www.swissadme.ch) [29]. Following, structural geometry optimization was performed for the selected flavanols by density functional theory (DFT) method with a hybrid functional Becke–Lee-Yang–Parr (B3LYP) supplemented with standard 6-31G** basis set in GAUSSIAN-03 suite [30]. The structural geometries for these compounds were fully optimized without any constraints at global minima and inclusion of imaginary frequency modes as reported earlier [31,32]. Next, the optimized geometries of compounds were then used in frontier molecular orbital calculations and molecular docking simulations with MMP-1 as reported earlier [31]. 

### 2.3. Molecular Docking Simulation

The 3D structures of flavanols optimized earlier by B3LYP/6-31G** were prepared using LigPrep module in Schrödinger suite for docking experiments [33]. Besides, crystal structure of the human MMP-1 was also processed using a multi-step process with Prime and protein preparation wizards in Schrödinger suite [33,34]. Briefly, the native ligand was removed from the receptor and water molecules were retained in the active pocket as recent studies elucidated the active role of water molecule in the peptide hydrolysis by the active form of MMP-1 [35]. Additionally, polar hydrogen atoms were added to the protein structure followed by structural refinement using protein preparation wizard. Here, charges and atom types corresponding to the right bond orders were allocated while side chains, which are not included in the formation of salt bridges, were neutralized. Additionally, hydrogen-bonding network optimization, rotation of thiol hydrogen and hydroxyl atoms, and generation of tautomerization, and protonation states of His residues as well as Chi “flip” assignments for residues Asn, Gln, and His residues were accomplished using the protassign script. Furthermore, a standard distance-dependent dielectric constant at 2.0 Å, accounting for electronic polarization and small backbone variations in the receptor, and a conjugated gradient algorithm were subjected to subsequent refinement of the receptor under optimized potentials for liquid simulations (OPLS)-2005 force field with the maximum root mean square deviation (RMSD) for the non-hydrogen atoms reached at 0.3Å [36,37]. Following, (-)-epicatechin, proanthocyanidin B2, and EGCG were docked at the binding pocket containing active residues, i.e., Gly179, Asn180, Leu181, Ala182, His183, Tyr210, Val215, His218, Glu219, His222, His228, Tyr237, Pro238, Ser239, and Tyr240 as well as metal ion Zinc275, as reported earlier in the crystal structure of MMP-1 with hydroxamate inhibitor [26]. Next, molecular docking simulation of the selected compounds in the active pocket of MMP-1 was conducted by extra precision (XP) protocol of GLIDE5.8, Schrödinger suite [34,38], where receptor was treated as rigid entity and ligands were subjected to change in conformation to attain the most feasible interaction profile with active residues. Also, hydrogen bond, van der Waals, polar interactions, metal binding, coulombic, freezing rotatable bonds, hydrophobic contacts, penalty for buried polar groups, water desolvation energy, and binding affinity enriching interactions were measured in Glide XP scoring protocol [32,39]. Finally, the best docked poses of ligands with the highest docking energy in the active pocket of MMP-1 were considered for intermolecular interaction analysis using Maestro tool in the Schrödinger suite [33,34].

To calculate the change in ligand conformation following molecular docking simulation, the selected best poses of the ligands were extracted from the docked complexes and aligned to respective B3LYP/6-31G** optimized initial geometries using Maestro tool in the Schrödinger suite [33,34].

### 2.4. Explicit Molecular Dynamics Simulations

The molecular docked complexes of MMP-1 with (-)-epicatechin, proanthocyanidin B2, and reference compound EGCG were further studied for complex stability with respect to time via 500 ns molecular dynamics (MD) simulation intervals under Linux environment on HP Z238 Microtower workstation using Desmond v4.4 module of Schrödinger-Maestro v10.4 [40,41]. Briefly, each protein–ligand complex was fabricated as an orthorhombic grid box (10Å × 10Å × 10Å buffer) and immersed in a Monte Carlo equilibrated periodic transferable intermolecular potential 4-point (TIP4P) water bath. Besides, salt, and counter ions were added to neutralize the system while placed at a distant of 20 Å from the ligand using system builder tools in Desmond-Maestro interface. Following initial equilibration under default conditions, a 0.002-ps time step was fixed for the anisotropic diagonal position scaling to contain the constant pressure during MD simulation. Furthermore, the temperature of the system was fixed at 300 K, associated with 20-ps normal pressure and temperature (NPT) reassembly at 1 atm pressure, and system density was kept near 1 g/cm^3^; other parameters with default setup were used for system preparation and in the MD simulation protocol. Finally, MD simulation for each complex was performed for 500-ns intervals under similar conditions. Later, generated MD simulation trajectories for each complex were analyzed by the simulation interaction diagram tool and simulation event analysis module in Desmond v4.4 module of Schrödinger-Maestro v10.4 suite [33,34].

### 2.5. Post Molecular Simulation Quantum Chemical Calculations

To calculate the change in the electronic properties of the docked bioactive compounds in the active pocket of MMP-1, hybrid QM/MM (quantum mechanics/molecular mechanics) calculations were performed on the molecular docked complexes and final snapshots extracted from the respective MD simulation using the ONIOM two-layer system in GAUSSIAN-03 suite [30,42]. Herein, the higher layer was considered for the docked ligand in the active pocket of proteins and was treated with the quantum mechanical method, i.e., the B3LYP/6-31G** method, while the protein was considered under the lower layer and treated with a universal force field (UFF) as molecular mechanics (MM) region. Finally, the frontier molecular orbitals, i.e., highest occupied molecular orbitals (HOMO), lowest unoccupied molecular orbitals (LUMO), and band energy gap, were calculated for both the docked poses and extracted snapshots from the MD simulation trajectory of the respective complexes.

### 2.6. Molecular Mechanics Generalized Born Surface Area (MM/GBSA) Calculations

The end-point binding free energy for the docked MMP-1-flavonols complexes and respective extracted snapshots from MD simulation were calculated using the Prime MMGBSA module of the MM/GBSA protocol in Schrödinger suite [33,34]. The formulas to calculate the end-point binding free energies and their respective individual decomposed energy components are mentioned in Equations (1)–(3), where the total binding free energy (ΔG_Bind_) is the sum of free energy difference between the docked-state complex (G_Com_) and the free-state individuals of the receptor and the ligand (G_Rec_ + G_Lig_). According to the second law of thermodynamics, the calculated Δ*G*_Bind_ for the protein–ligand complex can be further disintegrated into the enthalpy part (Δ*H*) and the entropy part (−*T*Δ*S*), as shown in Equation (1). However, most of the reported entropy computational methods, such as ranging from postprocessing approaches [43,44,45,46,47] to the simulation-synchronized methodologies [48,49,50] required hundreds of nanosecond simulation intervals to precisely calculate the entropies, and such calculations were tested only on a set of small molecules with only hundreds of atoms. These reported methods may be not suitable for the entropy calculations on more complex systems like drug-target complexes containing thousands of atoms; hence, entropy calculations were ignored for MMP-1-flavonol docked complexes due to the extremely expensive computational cost and relatively low prediction accuracy, as suggested earlier [51]. Under these conditions, enthalpy can be considered equal to the binding free energy for the complex and can be expressed as the addition of the molecular mechanical energy (ΔE_MM_) and the solvation free energy (ΔG_Sol_). Typically, ΔE_MM_ is an addition to the intramolecular energy (ΔE_Int_, which is further composed of the bond, angle, and dihedral energies of the system), the electrostatic energy (ΔE_Ele_), and the van er Waals interactions (ΔE_vdW_). Furthermore, ΔE_Int_ can be in particular canceled as single molecular trajectories were considered in the binding free energy calculations. Thus, MM/GBSA calculations were performed with OPLS-2005 force field with default parameters on the molecular docked complexes and 200 snapshots extracted from 500 ns MD simulation trajectories of respective complexes, which were initially refined by removal of explicit TIP4P water molecules and ions as reported earlier [52,53].
(1)$${\Delta G}_{\mathrm{Bind}}={\Delta G}_{\mathrm{Com}}-\left({\Delta G}_{Rec\;}+\;{\Delta G}_{Lig}\right)=\Delta H-T\Delta S\;\approx {\;\Delta E}_{\mathrm{MM}}+{\Delta G}_{\mathrm{sol}}-T\Delta S$$
(2)$${\Delta E}_{\mathrm{MM}}={\Delta E}_{\mathrm{Int}}+{\Delta E}_{\mathrm{Ele}\;}+{\Delta E}_{\mathrm{vdW}}$$
(3)$${\Delta G}_{\mathrm{Sol}}={\Delta G}_{\mathrm{Pol}}+{\Delta E}_{\mathrm{Nonpol}\;}$$

### 2.7. MMP-1 Zymography

In this study, the zymography method was used to predict the inhibition activity of the selected bioactive compounds against MMP-1. Briefly, the human fibroblast cells (Detroit551, KCLB, Jongno, Seoul, Korea) were initially cultured in Modified Eagle Medium (MEM, Welgene, Gyeongsan, Gyeongbuk, Korea) with 10% fetal bovine serum (FBS) (Welgene)) and penicillin (100 units /mL)/streptomycin (100 μg/mL) (Welgene)) under 5% CO_2_ environment in CO_2_ cell incubator (NU-4750G, NUAIRE, Plymouth, MN, USA) for 48 h. Following, 1 × 10^6^ cells were dispensed in 100 mm plate, fixed for 24 h, and cultured until the cells developed 80–90% confluence of the plate; later, the upper layer of culture solution was collected and centrifuged (10,000 rpm, 20 min, 4 °C) to remove cells from the culture medium. Following, for extraction of the protein from the collected culture medium, one volume of supernatant was gently mixed with 4 volumes of absolute chilled acetone (−20 °C), stored at −20 °C for 60 min, and then centrifuged (10,000 rpm, 20 min, 4 °C) to collect the precipitated total protein. After that, the collected protein was air dried at room temperature for 30 min and dissolved in Protein Extraction Reagent (ThermoFisher Scientific, Waltham, MA, USA) to dissolve the protein pellets followed by protein quantification using BCA^TM^ Protein Assay Kit (ThermoFisher Scientific) against different concentrations, as mentioned in the assay kit instruction manual. 

Next, 15 μg of the protein sample was incubated with various concentrations (0–1000 μg/mL) of selected bioactive compounds for 30 min at room temperature and later loaded in the 10% sodium dodecyl sulfate-polyacrylamide gel (SDS-PAGE) containing 0.1% casein (Sigma-Aldrich, St. Louis, MO, USA) as a substrate for MMP-1 zymography. After electrophoresis, the gel was washed twice with a washing buffer (2.5% Triton X-100, 50 mM Tris-HCl (pH 7.5), 5 mM CaCl_2_, and 1 μM ZnCl_2_) at room temperature for 30 min, followed by incubation in a buffer ((1% Triton X)-100, 50 mM Tris–HCl (pH 7.5), 5 mM CaCl_2_, 1 μM ZnCl_2_) at 37 °C for 24 h for casein hydrolysis by MMP-1. The digested gel was then stained with 0.3% Coomassie brilliant blue R-250 (BIO BASIC, Amherst, NY, USA) for 2 h followed by incubation in detaining solution (methanol: acetic acid: water = 50:10:40) for 10 min. Finally, casein degradation by MMP-1 activity was measured through the size of the transparent band from three independent experiments under similar experimental conditions. The band intensity that appeared transparently on casein zymography was expressed as the MMP-1 activity graph by measuring the area of the band using the LabWorks (UVP, Upland, CA, USA) program. This protocol was applied for each compound, i.e., (-)-epicatechin, proanthocyanidin B2, and reference compound EGCG, at various concentrations (0–1000 μg/mL) to monitor the inhibition of MMP-1 activity against casein as a substrate.

### 2.8. Western Blot Analysis for MMP-1

To validate the position of MMP-1 in the zymographs, western blotting was performed for the protein collected from cell medium and cell extract of the human fibroblast (Detroit551) cells by comparison to the human keratinocyte (HaCaT) cells. Briefly, the human fibroblast (Detroit551, KCLB, Jongno, Seoul, Korea) cells were inoculated in Modified Eagle Medium (MEM, Welgene, Gyeongsan, Gyeongbuk, Korea) medium while the human keratinocyte (HaCaT, ATCC, Manassas, VA, USA) cells were inoculated in Dulbecco’s Modified Eagle Medium (DMEM, Welgene, Gyeongsan, Gyeongbuk, Korea), amended with 10% fetal bovine serum (FBS) (Welgene, Gyeongsan, Gyeongbuk, Korea) and penicillin (100 units/mL)/streptomycin (100 μg/mL) (Welgene, Gyeongsan, Gyeongbuk, Korea), and followed by incubation under 5% CO_2_ in CO_2_ cell incubator (NU-4750G, NUAIRE, Plymouth, MN, USA for 48 h).

Following, grown cells were collected and lysed using lysis buffer (M-PER ^®^ Mammalian Protein Extraction Reagent) (ThermoFisher Scientific, Waltham, MA, USA) followed by centrifugation at 12,000 rpm for 10 min. Next, total protein concentration was estimated in each solution as explain earlier in Section 2.7. Then, for western blotting, 2 μg/μL of protein was dissolved in sample buffer (62.5 mM Tris-HCl (pH 6.8), 10% glycerol, 1% SDS, 1% *β*-mercaptoethanol, and 0.001% bromophenol blue) and denatured by boiling over water bath for 3 min. After that, an aliquot of denatured protein containing 30 μg of protein was electrophoresed in a 7.5% SDS-PAGE. Later, this gel was transferred onto a polyvinylidene fluoride (PVDF) membranes (Millipore Corp., Bedford, MA, USA) and incubated in TBS-T blocking buffer (100mM Tris-HCL, 0.9% NaCl, and 0.05% Tween-20) for 20 min at room temperature; the membrane was then exposed to the antibodies (diluted 1/500: a rabbit anti-MMP-1 polyclonal antibody, procured from GeneTex, Irvine, CA, USA) in Western Enhanced buffer (NeoScience, Geumcheon, Seoul, Korea) overnight at room temperature. Later, the membrane was washed with TBS-T buffer, treated with horseradish peroxidase-conjugated antibodies (GeneTex, Irvine, CA, USA) for 2 h at room temperature, and followed by washing with TBS-T buffer. Finally, the western blots were developed for MMP-1 using enhanced chemiluminescence (ECL) kit (ThermoFisher scientific, Waltham, MA, USA).

## 3. Results and Discussion

### 3.1. ADME and Quantum Chemical Calculation Analysis

Initially, multiple computational methods were employed to predict the basic chemoinformatics and molecular properties of (-)-epicatechin and proanthocyanidin B2 by comparison to reference compound EGCG (Figure 1 and Appendix A). These intrinsic properties of the molecules along with pharmacological and drug-like properties are essentially required for medical applications [54,55]. In this study, drug-likeness of selected bioactive compounds was studied by calculating the ADME properties using the SwissADME, sever which revealed only (-)-epicatechin to follow all the drug-likeness rule (Appendix A). However, it should also be noted that drug-likeness rules do not apply to natural products or bioactive molecules which are recognized by an active transport system when considering “druggable chemical entities” [56,57]. Moreover, Koehn in 2012 summarized the 26 drugs approved between 1981 and 2011 derived from 18 natural products known to challenge the “rule of 5” and its structures [58]. Besides, based on the presence of electronegative atom numbers in the structure, each bioactive compound was also studied for the number of hydrogen-bond donors and acceptors atoms. Here, a minimum of 6 acceptor and 5 donor atoms were counted in (-)-epicatechin while a maximum of 12 acceptor and 10 donor atoms was listed in proanthocyanidin B2 (Appendix A). Also, other properties such as Lipophilicity (LIPO), size, polarity (POLAR), insolubility in water (INSOLU), instauration (INSATU), and flexibility (FLEX) were calculated for the bioactive compounds, as shown in Figure 1; in addition to other essential medical properties (Appendix A). All these properties suggested the ideal medicinal properties for (-)-epicatechin and proanthocyanidin B2 by comparison to reference compound EGCG. 

Furthermore, structural parameters such as bond length, bond angles, and dihedral angles were calculated for (-)-epicatechin, proanthocyanidin B2, and EGCG using B3LYP DFT calculations in conjunction with the 6-31G** basis set (Appendix A). Figure 2a,b depicts the optimized structural geometries without hydrogen atoms for (-)-epicatechin, proanthocyanidin B2, and EGCG generated by GaussView 3.0.8. These optimized structures of each compound were further studied for the electrostatic potential, which is generally used to differentiate the relative polarity on the molecule by identification of nucleophilic and electrophilic sites on the molecular surface [59]. Besides, electrostatic potential for the molecule has been suggested along with dipole moment to determine the several intermolecular interaction properties and the most favorable regions between ligand and receptor [60]. Figure 2 exhibits the electrostatic map predicted for (-)-epicatechin, proanthocyanidin B2, and EGCG based on electron density distribution on the respective molecular surface. It assisted in the estimation of molecular size, shape, and polarity, i.e., positive, negative, and neutral electrostatic potential sites over the molecule via the color grading scheme, where the blue color represents the most electropositivity, i.e., low electron region, and the red color stands for the most electronegative center or electron rich regions [61]. Moreover, color intensity on the electrostatic map in relation to energy (in Hartree) demonstrates electron density at a given point on the surface of molecule. In this study, all molecules were logged for blue color region on the electrostatic map, which indicates the strong electrophilic region produced by electrons in hydrogen atoms (Figure 2). Likewise, electrostatic maps exhibited a red color corresponding to the higher nucleophilic regions contributed by electron-rich species, such as hydroxy and carbonyl groups in the respective bioactive compounds. In addition, the color-coding regions of orange, yellow, and green in the electrostatic maps demonstrates the electronic transition within the respective molecule (Figure 2). Hence, the calculated electrostatic potential map for designated bioactive compounds suggested that the highly electronegative atoms, i.e., oxygen and electropositive atoms such as carbon atoms bonded with oxygen in cyclic chains and hydrogen atoms, have the capability to introduced intermolecular interactions with the active residues of proteins in their vicinity during molecular docking simulation.

According to the theory of frontier molecular orbitals, interaction and overlapping of two chemical species result in establishment of one or more molecular orbitals carrying lower bonding and higher antibonding energy [62]. Hence, energy contained in the produced orbitals, i.e., highest occupied molecular orbitals (HOMO) and lowest unoccupied molecular orbitals (LUMO) supplemented with attractive force of electrons, can be utilized to connect the electron donation and acceptor capacity, respectively for a chemical species [63]. Therefore, optimized geometries of the compounds, i.e., (-)-epicatechin and proanthocyanidin B2 along with reference compound EGCG, were predicted for the frontier molecular orbitals. Figure 3 shows a well-defined distribution of HOMO and LUMO over the two distinct parts of the molecules, where red and green color distributions marked negative and positive wave functions of the respective molecular orbitals. For instance, HOMO was predicted on the 3,4-dihydroxyphenyl ring, while 3,4-dihydro-2H-chromene-3,5,7-triol ring was notice for distribution of LUMO in the (-)-epicatechin molecule (Figure 3a). Likewise, HUMO and LUMO were observed on each dihydroxyphenyl ring of the (-)-epicatechin subunits in the proanthocyanidin B2 separated by a bond at positions 4 and 8′ in a β-configuration (Figure 3b). Also, in the reference compound EGCG, HOMO was distributed on the atoms of 3,4,5-trihydroxybenzoate ring while LUMO was located on the atoms of 3,4,5-trihydroxyphenyl ring in the molecular structure (Figure 3c). Also, the energy gap, i.e., HOMO–LUMO band gap energy (ΔE = E_HOMO_ − E_LUMO_), is generally used in quantum chemical calculations to predict the chemical reactivity of the compounds [64,65]. Herein, (-)-epicatechin and proanthocyanidin B2 exhibited higher band gap energies of 5.37 and 5.30 eV, respectively, against EGCG (4.42 eV) (Figure 3). Previously, frontier molecular orbital band gap energy calculated for the bioactive compounds Ganoderiol D and Ganodermanontriol were also observed with energy values above 4 eV [31]. It was established that a higher frontier orbital energy gap corresponds to higher kinetic stability and low chemical reactivity by comparison to lower counterparts, attributed by energetically resistance to the addition of electrons in a high lying LUMO from low-level HOMO [66]. Hence, the calculated quantum chemical properties suggested the higher stability and nonreactive chemical properties for the bioactive compounds (-)-epicatechin and proanthocyanidin B2 against reference compound EGCG.

### 3.2. Molecular Docking Simulation and Intermolecular Interaction Analysis

Molecular docking simulation plays a vital role in computer-aided drug design. This study helps to identify the potential small molecules via docking towards the binding pocket of the protein. The extra precision docking algorithm in the GLIDE module assists with the prediction of ligand binding affinities in the protein active site by including water desolvation energy, hydrophobic interactions between the ligand atoms and protein active residues, formation of neutral–neutral single or correlated hydrogen bonds under hydrophobically enclosed environment, and another five clusters of charged–charged hydrogen bond formation [39]. Among the various poses generated for the docked ligands in the active site of MMP-1 protein, the best poses with least docking RMSD and highest docking energy were selected for further intermolecular interaction analysis. The primary purpose of the study was to differentiate the most stable conformation of the docked ligands based on their score value which can inhibit the target protein [67]. For the MMP-1 protein, the selected bioactive compounds, viz. (-)-epicatechin and proanthocyanidin B2 along with reference compound EGCG, were docked on the essential residues of MMP-1 and respective binding energies were calculated using XP docking protocol (Figure 4).

Moreover, electrostatic interactions like in heteroatom-hydrogen bonds X-H···Y (X = O or N; Y = O, N, or halogen), salt bridges, van der Waals interactions (π-π interactions), carbon–hydrogen, and metal interactions are required for the formation and stability of the protein–ligand complexes [68,69,70]. Especially, hydrogen bond is observed as an intermolecular interaction that exhibits covalent, electrostatic, and van der Waals properties [71,72] Therefore, hydrogen bonds not only mediate protein–ligand binding but also influence the physicochemical properties of the molecules, such as solubility, distribution, partitioning, and permeability, which are key characteristics for drug development [73,74]. Typically, in protein–ligand complex, hydrogen bond formation such as C = O···H, N–H···O, O–H···O, and N–H···N are classified as strong bonds because of presence of highly electronegative atoms, i.e., Nitrogen (N) and Oxygen (O), while weak bonds such as C–H···O and C–H···N are categorized as the most common type of intermolecular hydrogen bond formation between the protein and ligand [75]. Also, strong hydrogen bonds (X-H···Y) are well known to have shorter bond lengths than the sum of van der Waals radii of heteroatoms X and Y [76]. Thus, the hydrogen bonds formation between active residues and ligand were identified using the relaxed criteria, as established by McDonald and Thornton [77], of donor–acceptor separation ≤ 3.5 Å, hydrogen–acceptor distance ≤ 2.5 Å, and hydrogen bond angle ≥ 60°. Infrequently, some buried donor and acceptor atoms formed hydrogen bonds with interstitial water molecules at the docked site of ligand with protein. To permit disorder contributed by solvent molecules, the criteria for hydrogen bonding was further relaxed for donor–acceptor distance at a cutoff value of 4.5 Å only in presence of water molecule. Furthermore, Bissantz, Kuhn, & Stahl, 2010, et al., exhibited that the mean donor–acceptor distance of a typical hydrogen bond in protein–ligand complex exists in a range between 2.8–3.1 Å with a donor hydrogen-acceptor angle about at 130° [78]. Hence, based on the literature, the hydrogen bond between donor and acceptor of distances at approximately 2.2–2.5 Å, 2.5–3.2 Å, and 3.2–4 Å were marked as strong with covalent character, with moderate electrostatic features, and with week electrostatic interactions, respectively. With these criteria under consideration, the collected best docked poses of MMP-1 with selected bioactive compounds were analyzed for formation of hydrogen bond and other intermolecular interactions within a distance of 4 Å as the default parameter in the Maestro-Schrödinger suite shown in Figure 4.

The docked complex of MMP-1–(-)-epicatechin showed a binding affinity of −7.862 kcal/mol at an RMSD of 0.001 and formed four strong hydrogen bonds, i.e., two hydrogen bonds by hydroxy and oxy groups in 3,4-dihydro-2H-chromene-3,5,7-triol rings with Gly179 (donor hydrogen bond C = O···H; 1.75 Å) and Tyr240 (acceptor hydrogen bond N–H···O; 2.45 Å) residues and two donor hydrogen bond formation by hydroxy functional group in 3,4-dihydroxyphenyl ring with Glu219 (H C–O···H; 1.60 Å and C–O···H; 2.15 Å); additionally, one donor hydrogen bond from the hydroxy functional group in 3,4-dihydro-2H-chromene-3,5,7-triol rings mediated by water molecule with residue Tyr210 at a 4.03 Å distance was also recorded (Figure 4a,b). Besides, hydrophobic (Leu181, Ala182, Tyr210, Val215, Tyr237, Pro238, and Tyr240), polar (Asn180, His218, and Ser239), negative (Glu219), Glycine (Gly179), and positive (Arg214) interactions were also recorded in an MMP-1–(-)-epicatechin docked complex (Figure 4a,b).

Likewise, MMP-1–proanthocyanidin B2 complex shows docking score of −8.145 kcal/mol at a 0.007 RMSD with a significant intermolecular interaction profile exhibiting four hydrogen bond formation, viz. hydroxy groups in 3,4-dihydroxyphenyl ring exhibited two donor hydrogen bond formations with residue Tyr237 (C–O···H; 2.11Å and C–O···H; 2.14Å) while the hydroxy and oxy groups in 3,4-dihydro-2H-chromene-3,5,7-triol ring were observed for formation of acceptor hydrogen bond with Ser239 (O–H···O; 1.99Å) and donor hydrogen bond with Tyr240 (C = O···H; 1.59Å) residues, respectively. Also, single donor and acceptor hydrogen bond formation was exhibited by one hydroxy group on a 3,4-dihydro-2H-chromene-3,5,7-triol ring mediated by two water molecules for residues Gly179, Glu209, and Tyr210 at distances of 5.04, 5.09, and 5.10 Å, respectively (Figure 4c,d). Moreover, hydrophobic (Leu181, Tyr210, Ile232, Tyr237, Pro238, and Tyr240), polar (Asn180, Ser239, and Thr241), negative (Glu209), and glycine (Gly179) interactions were also observed in the respective docked complex (Figure 4c,d).

Whilst the docked complex of MMP-1 with reference compound EGCG showed a docking energy of −6.488 kcal/mol with 0.0013 RMSD and a considerable number of hydrogen bonds, and in addition to other intermolecular interactions with the active pocket residues of MMP-1 (Figure 4e,f). Herein, a total of two donor hydrogen bond formations by hydroxy group in 3,4,5-trihydroxybenzoate ring were recorded with the active residue Glu219 (C–O···H; 1.52Å and C–O···H; 1.69Å) while the oxy group in the same ring exhibited acceptor hydrogen bond with residue Tyr240 (N–H···O; 2.73Å). Also, two donor hydrogen bonds by the hydroxy group on the 3,4,5-trihydroxyphenyl ring in the EGCG molecule were observed with Tyr210 residue at a distance of 4.10 Å (Figure 4e,f). Interestingly, catalytic Zn275 was also observed within a 4-Å area around the docked EGCG in the active pocket of MMP-1. Besides, hydrophobic (Leu181, Ala182, Tyr210, Val215, Pro238, and Tyr240), polar (Asn180, His183, His218, His222, and His239), negative (Glu219), and glycine (Gly179) interactions were recorded in the MMP-1–EGCG docked complex (Figure 4e,f). Remarkably, atoms in the respective aromatic rings in all three bioactive compounds forming donor and acceptor hydrogen bonds with active residues of MMP-1 were initially predicted for HOMO and LUMO distribution, respectively (Figure 3 and Figure 4)

Based on the docking score, (-)-epicatechin and proanthocyanidin B2 were predicted as potential inhibitors of MMP-1 by comparison to reference compound EGCG. This observation was also supported by substantial intermolecular interactions with the active and substrate binding residues of MMP-1 in the respective docked complexes. These interacting residues, i.e., Gly179, Asn180, Leu181, Ala182, His183, Tyr210, Val215, His218, Glu219, His222, His228, Tyr237, Pro238, Ser239, and Tyr240, in this study were the same as reported in the crystal structure of MMP-1 (PDB ID: IHFC) with ligand methylamino–phenylalanyl–leucyl–hydroxamic acid [26]. Interestingly, residues His218, His222, and His228 identified a Zinc 2 metal ion binding site and catalytic cleft in the crystal structure of MMP-1 which extends past the zinc ion alongside the D-strand of the protein structure [26]. Also, residue Glu219 as an active site was elucidated to promote the nucleophilic attack of water molecules on the scissile peptide bond of the substrate whilst the side chain oxygen on the residue Asn180 (Calcium 3 calcium binding site via carbonyl oxygen) and carbonyl oxygen of residue Ala182 were predicted to maintained the complex of protonated nitrogen of the scissile bond; in addition, residues Gly179, Leu181, Pro238, and Tyr240 were defined as substrate binding sites in the catalytic pocket of MMP-1 [26]. Moreover, His183 (Zinc 1, metal binding residue) was identified as an additional active residue in the crystal structure of MMP-1 [26,79].

It is important to mentioned that only (-)-epicatechin and EGCG showed two and one hydrogen bond formations, respectively with the active residue (Glu219); in addition, interactions with substrate binding residues were also logged in the docked complexes (Figure 4). However, proanthocyanidin B2 exhibits only three hydrogen bond formations with the substrate binding residues, viz. Tyr237, Ser239, and Tyr240 (Figure 4). These results support that (-)-epicatechin can directly blocks active residues as in reference compound EGCG while proanthocyanidin B2 inhibits the enzyme activity by occupying the substrate binding region in the protein structure. It was reported that the molecular mechanism for inhibition of MMP-1 may involve direct competition with the substrate by establishing hydrophobic interactions and hydrogen bond formation with the backbone amide groups and other side chain functional groups of MMP-1 that results in significant conformation change in the protein structure [80,81].

Further, the docked poses of bioactive compounds, i.e., (-)-epicatechin, proanthocyanidin B2, and EGCG, were also superimposed on the initial B3LYP/6-31G** optimized structural geometries, revealing no considerable changes in the ligand conformation after molecular docking, except a degree of conformational rotation was observed in the dihydroxyphenyl ring of each bioactive compound (Appendix A). These structural rotations in the ring were suggested due to consideration of rotation in hydrogen bond opted in the GLIDE docking protocol to produce the best pose of the ligand for substantial intermolecular interactions with the active pocket of MMP-1 (Appendix A and Figure 4).

### 3.3. Molecular Dynamics Simulation Analysis

An interaction profile predicted from on target-ligand calculated using the molecular docking approach can be further validated by molecular dynamics simulation and crystallographic studies [82,83,84]. Herein, molecular dynamics (MD) simulation was conducted to find out the stability, conformation, and intermolecular interactions between the ligands and active residues of the MMP-1 protein over a 500-ns interval using the Desmond package. The generated MD simulation trajectories of each complex were subjected to specific parameters such as last pose analysis, root mean square deviation (RMSD), root mean square fluctuation (RMSF), and protein–ligand interaction profile to understand the bonding nature of the respective ligand at the active pocket of protein during simulation interval.

Initially, intermolecular interaction analysis for the last poses extracted from simulation trajectory was studied for respective complex stability (Figure 5). Remarkably, all complexes showed the presence of the ligand at the active pocket of MMP-1 and strong intermolecular interaction such as hydrogen bond formation, hydrophobic, polar, negative, positive, and glycine interactions (Figure 5). For instance, in the MMP-1–(-)-epicatechin complex, hydroxy groups on the 3,4-dihydro-2H-chromene-3,5,7-triol ring exhibited strong and moderate donor hydrogen bonds with residues Ala182 (C = O···H; 1.80Å) and Glu219 (C–O···H; 2.72 Å), respectively, while one moderate acceptor hydrogen bond was monitored in the same aromatic ring with residue Leu181 (N–H···O; 2.72 Å). Likewise, the extracted pose for the MMP-1–proanthocyanidin B2 complex showed one strong donor hydrogen bond with residue Asn180 (C = O···H; 1.92 Å) along with π–π interactions at His218 and His228 residues. Similarly, analysis for MMP-1–EGCG complex revealed formation of two strong donor hydrogen bonds through the hydroxy group on the 3,4,5-trihydroxybenzoate ring with active residue Glu219 (C–O···H; 1.55 Å and C–O···H; 1.78 Å) and one strong donor hydrogen bond by the hydroxy group on the 3,4,5-trihydroxyphenyl ring at residue Gly179 (C = O···H; 2.11Å) (Figure 5). These interacting residues in the respective protein–ligands docked complex were also logged in the initial molecular docked poses (Figure 4), indicating the considerable stability of respective complex during a 500-ns MD simulation interval.

Besides, alpha carbon (Cα) deviations were also calculated for MMP-1 from the RMSD value during a 500-ns MD simulation interval (Figure 6). All complexes showed stable variations <2 Å during simulation interval with average values of 1.85, 1.88, and 1.81 Å for (-)-epicatechin, proanthocyanidin B2, and EGCG docked complexes with MMP-1, respectively (Figure 6). These results suggested no significant structural change in MMP-1 protein structure bound with respective ligands during the MD simulation. Moreover, RMSD values for the protein fit ligands exhibited deviations within 100 ns and followed by state of equilibrium till a 500-ns interval (Figure 6). These observations suggested the presence of the ligand at the active pocket of MMP-1 during simulation. Besides, averages of considerable RMSD values of 6.08, 11.62 and 3.72 Å were logged for the (-)-epicatechin, proanthocyanidin B2, and EGCG docked on MMP-1, respectively (Figure 6). These considerable RMSDs in the protein structure and ligands were further supported by acceptable RMSF values for residues of protein (<3.5 Å) and atoms in ligands (<4.5 Å), respectively, during the MD simulation (Appendix A). These calculated respective RMSD and RMSF values suggested the stability of docked complexes of MMP-1 with (-)-epicatechin, proanthocyanidin B2, and EGCG during a 500-ns simulation interval.

Moreover, atomic level information during simulation interval is essential to calculate the binding pose of ligands with the active site of protein. For binding mode analysis, the intermolecular interactions such as hydrogen bond, hydrophobic contact, ionic interaction, and salt bridge formation were analyzed over a 500 ns MD simulation interval (Figure 7 and Appendix A). Particularly, a number and specific type of hydrogen formation between the protein and docked ligands were also extracted from the respective MD simulation trajectories. Figure 7 shows a higher number of hydrogen bond formations in the MMP-1–EGCG (average 4 bonds) against MMP-1–(-)-epicatechin (average 2 bonds) and MMP-1–proanthocyanidin B2 (average 1 bonds). Moreover, acceptor hydrogen bonds were found prominent among the hydrogen bond types formed between selected bioactive compounds and MMP-1 active residues; (-)-epicatechin and EGCG exhibited considerable hydrogen bond formation with active Glu219, while proanthocyanidin B2 exhibited hydrogen bond with substrate binding residue Asn180. Additionally, a total intermolecular fraction analysis for the docked bioactive compounds with MMP-1 during simulation revealed significant contacts with the essential residues, i.e., His218, Glu219, His222, and His228 (Appendix A) in addition to other active pocket residues of MMP-1 as observed in the respective docked complexes (Figure 4). Interestingly, only proanthocyanidin B2 exhibits substantial ionic interactions with residues His218, His222, and His228 during a 500 ns MD simulation interval (Appendix A). These observations suggested significant protein–ligand intermolecular interactions along with substantial stability of (-)-epicatechin and proanthocyanidin B2 by comparison with EGCG on the active pocket of MMP-1 throughout 500 ns MD simulations; hence, indicated the inhibitory potential of selected bioactive compounds, i.e., (-)-epicatechin and proanthocyanidin B2 against human MMP-1.

### 3.4. Post-Molecular Simulation Quantum Chemical Calculations

A two-layer ONIOM method was selected for hybrid QM/MM calculations to calculate the HOMO and LUMO for the molecular docked MMP-1–bioactive compound complexes along with respective last snapshot extracted from 500 ns MD simulation. Figure 8 shows HOMO and LUMO for the bioactive compounds, i.e., (-)-epicatechin, proanthocyanidin B2, and EGCG generated under the interaction influence imposed by the residues in the active pocket of MMP-1. By comparison, docked ligands with the initial optimized ligands showed no momentous change in HOMO and LUMO distribution, except a reduction in size of the wave function for all ligands docked with proteins was observed (Figure 8). Moreover, a notable change in orbital energy was also logged, which results in a change in energy band gap compared to the respective values calculated for the initial geometry of a ligand using a set of DFT calculations, suggesting the influence of intermolecular interactions formed during docking simulation on the respective ligand. These observations were further supported by the remarkable change in energy band gap for the docked bioactive compounds compared to respective optimized geometries and indicate induction of electron transfer within the ligand molecule to support significant intermolecular interactions in the docked complex. These statements can be supported by donor and acceptor hydrogen bond formation in the respective docked complexes (Figure 4). Furthermore, in comparison, frontier orbital energy and band gap energy of the docked bioactive compounds with respective complexes after 500 ns MD simulation exhibited no substantial differences; these results suggested the stable interaction between the active residues and docked bioactive compounds (Appendix A). Based on hybrid QM/MM calculations, the selected bioactive compounds, i.e., (-)-epicatechin, proanthocyanidin B2, and EGCG, were suggested with considerable ADME and drug-likeness and can be used for further in vitro evaluation.

### 3.5. Binding Free Energy Calculations

Typically, binding affinity is marked for the Gibbs energy of binding (ΔG), which is the result of two different terms, i.e., enthalpy (ΔH) and entropy (TΔS), and high affinity is only achievable when both energy components contribute favorably to the binding process [85,86]. However, several studies have shown difficulties in calculating simultaneous optimization of enthalpy and entropy [87]. Generally, two different forms of forces contributed in the binding of a drug molecule in the active pocket of its receptor, i.e., attractive forces such as hydrogen bonding and van der Waals interactions, and repulsive forces, including a hydrophobic effect that tends to move the drug molecule into hydrophobic cavity from the aqueous solvent [87]. As these forces differentially influenced the thermodynamic signature such as enthalpy and entropy contribution in drug binding with receptor [88,89], these factors have been suggested to provide an exclusive experimental way to characterize the binding mode of the drug. Recent studies suggested that a favorable interaction enthalpy is indicative of good interactions of drug molecule with the receptor [87]. Moreover, dispersion energy corrections also have a small influence on the optimized geometries of ligands and, in turn, binding pose with the protein; however, such dispersion energy calculations are suggested to be important in modeling reactions catalyzed by enzymes [90].

In this study, to compare how MD simulation affects the binding of ligands with active residues in the active pocket of MMP-1, binding affinity of the docked complexes, viz. MMP-1–(-)-epicatechin, MMP-1–proanthocyanidin B2, and MMP-1–EGCG, were calculated before and after MD simulation using the Prime MM/GBSA method (Figure 9). A list for ΔG_Bind_ and individual energy components calculated for the respective docked complexes before and after MD simulation are shown in Appendix A. Among the bioactive compounds, MMP-1–(-)-epicatechin and MMP-1–EGCG docked complexes showed 2.35 and 1.40 times increases in binding free energy, respectively, following MD simulation. These observations support the favorable attraction enthalpy contributed by hydrogen bonding and electrostatic interactions in the respective complexes after simulation. However, MMP-1–proanthocyanidin B2 complex exhibited 1.92 times decrease in the free binding energy after 500-ns simulation (Figure 9, Appendix A), marked for enhanced unfavorable interactions in the complex during simulation via a hydrophobic effect. Additionally, analysis of individual components contributing in the total binding free energy of the respective complexes, i.e., ΔG_Bind Coulomb_, ΔG_Bind Covalent_, ΔG_Bind vdW_ (van der Waals forces), ΔG_Bind Solv SA_ (solvent accessible Surface Area), and ΔG_Bind Solv GB_ (solvation energy Generalized Born), revealed significant contribution of ΔG_Bind Coulomb_ and ΔG_Bind vdW_ energy in the respective complex stability (Figure 8, Appendix A), supporting favorable enthalpy formation in the respective complexes. Similar results for the energy components ΔG_Bind Coulomb_ and ΔG_Bind vdW_ contributing in the docked complexes were reported earlier [91]. Therefore, it was suggested that (-)-epicatechin has relatively stronger affinity with the active pocket of MMP-1 than proanthocyanidin by comparison to reference compound EGCG, as predicted from molecular docking and MD simulation analysis (Figure 4 and Figure 5); and hence, (-)-epicatechin and proanthocyanidin were predicted as strong and weak inhibitors of MMP-1, respectively against EGCG, as concluded from binding affinity (MM/GBSA) calculations.

### 3.6. MMP-1–Zymography Analysis

As, it is well known that activated MMP-1 is released into the culture medium by the cells, the collected protein from medium was incubated with various concentrations of selected compounds and MMP-1 zymography was performed in the presence of beta casein as the substrate. Remarkably, only (-)-epicatechin and EGCG exhibited concentration-dependent inhibition of MMP-1 with maximums of 69.2% and 75.6% MMP-1 inhibition at 1000 μg/mL (Figure 10). However, proanthocyanidin B2 had a maximum inhibition of 43.4% at the highest concentration of 1000 μg/mL (Figure 10). Also, this band at 54 kDa in the zymography was established for the human MMP-1 using western blotting (Figure 10). Furthermore, these observations were in accordance with the respective molecular simulation analysis and binding free energy calculation which showed strong stability and affinity for (-)-epicatechin against proanthocyanidin B2 by comparison to reference compound EGCG (Figure 4, Figure 5, Figure 6, Figure 7, Figure 8 and Figure 9). These observations suggested the strong anti-proteolytic MMP-1 activity of monomeric (-)-epicatechin against dimeric (-)-epicatechin, i.e., proanthocyanidin B2.

## 4. Conclusions

Plant secondary metabolites are well known to quench the free radical generation and to inhibit many drug-target proteins; in addition, the bioavailability of plant secondary metabolites such as flavanols has been evaluated in humans. In this study, the interaction of the selected monomeric flavanol, i.e., (-)-epicatechin against dimeric flavanol, viz. proanthocyanidin B2, was computed using computational and validated using zymography. Initial ADME and quantum calculations along with molecular docking established the drug-likeness and substantial affinity at the active pocket of MMP-1 for (-)-epicatechin compared to proanthocyanidin B2. Also, molecular dynamics simulation and free binding energy calculations exhibited the contribution of hydrogen bonding and electrostatic interactions in the stability of (-)-epicatechin with active residues against proanthocyanidin B2, which can be clearly observed from the MMP-1 zymography gels. Therefore, the findings of this study advised (-)-epicatechin as a druglike molecule which could be further extrapolated to clinical/in vivo studies for the development of antiaging care products or therapeutics to treat chronic inflammatory diseases caused by rapid degradation of collagenase by MMP-1.

## Figures and Tables

**Figure 1 biomolecules-10-01379-f001:**
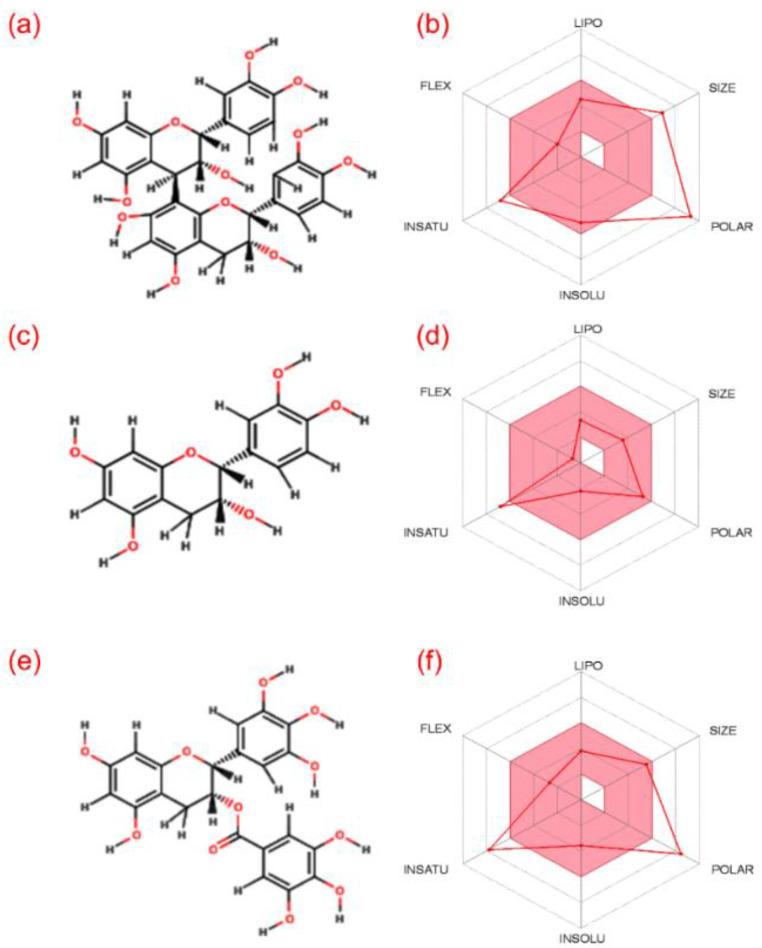
ADME analysis for the selected bioactive compounds using SwissADME sever, i.e., (**a**,**b**) (-)-epicatechin, (**c**,**d**) proanthocyanidin B2, and (**e**,**f**) reference compound epigallocatechin gallate (EGCG): here, the left panel shows the 2D structural formula used for calculating the properties on the SwissADME server, which are depicted on right panel, viz. Lipophilicity (LIPO), size, polarity (POLAR), insolubility in water (INSOLU), instauration (INSATU), and flexibility (FLEX).

**Figure 2 biomolecules-10-01379-f002:**
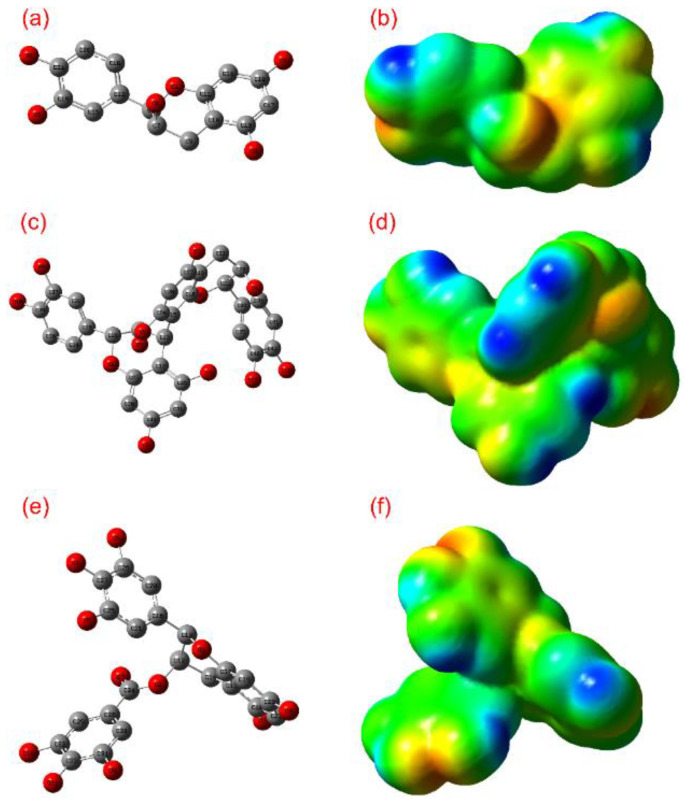
Optimized geometries and electrostatic potential map generated for (**a**,**b**) (-)-epicatechin, (**c**,**d**) proanthocyanidin B2, and (**e**,**f**) EGCG calculated using the density functional theory (DFT) B3LYP/6-31G** method.

**Figure 3 biomolecules-10-01379-f003:**
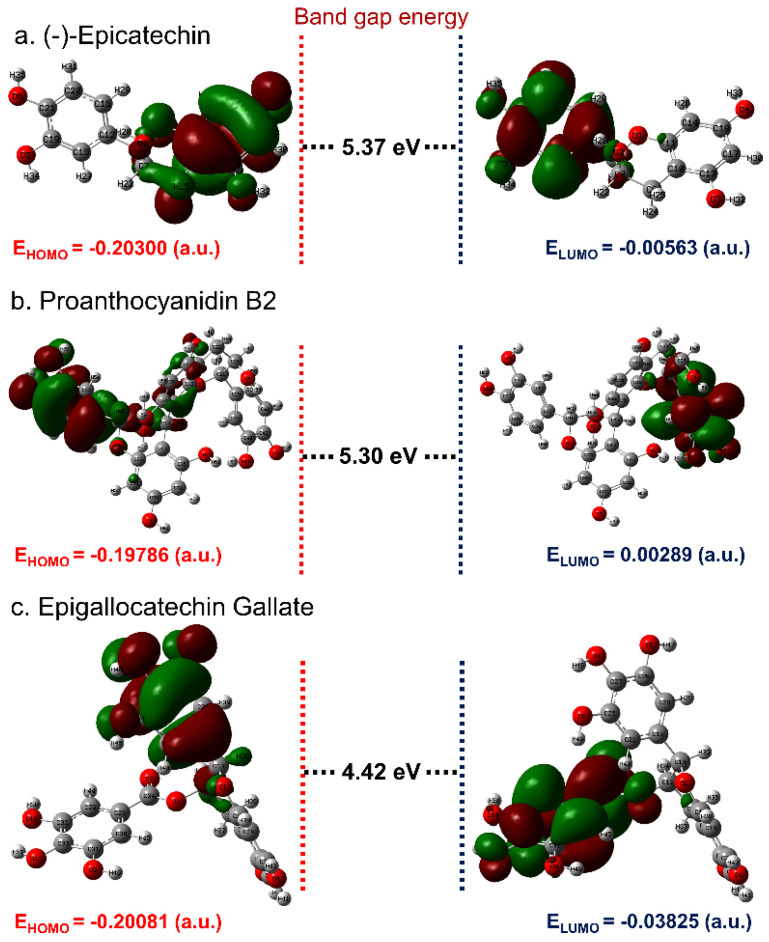
Frontier molecular orbital distribution on the optimized molecular structure of bioactive compounds, viz (**a**) (-)-epicatechin, (**b**) proanthocyanidin B2, and (**c**) EGCG along with predicted orbital energy and band gap energy values calculated using the DFT B3LYP/6-31G** method.

**Figure 4 biomolecules-10-01379-f004:**
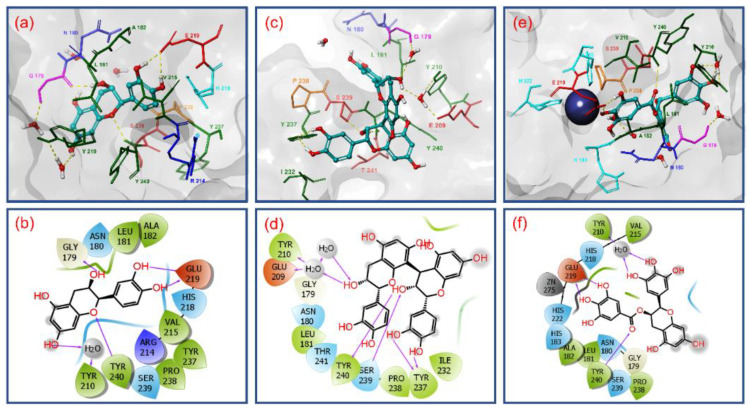
Three-dimensional and two-dimensional interaction mapping of selected ligands, i.e., (**a**,**b**) (-)-epicatechin, (**c**,**d**) proanthocyanidin B2, and (**e**,**f**) EGCG monitored within 4 Å space around the ligand in the active pocket of Matrix metalloproteinases 1 (MMP-1): in the 2D interaction map, pink arrows represent the hydrogen bonds while green, blue, red, violet, and grey color residues stand for the hydrophobic, polar, negative, positive, and glycine interactions, respectively, in the docked complexes.

**Figure 5 biomolecules-10-01379-f005:**
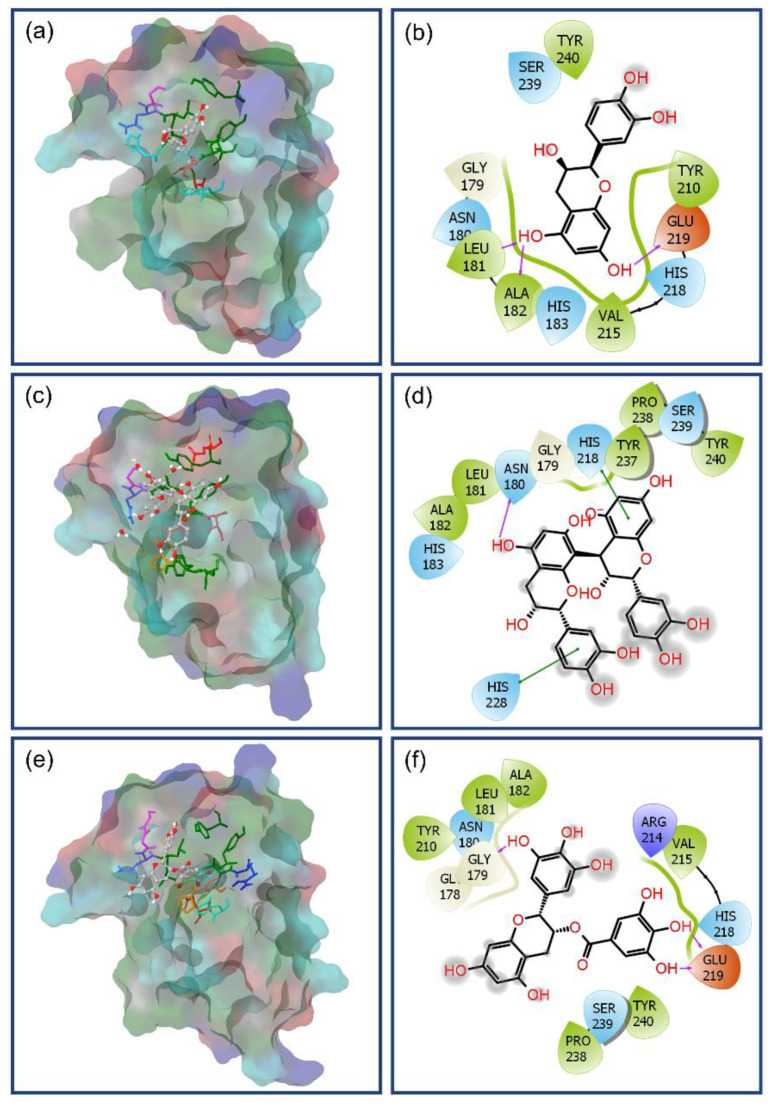
Last poses collected from 500 ns molecular dynamics (MD) simulation of (**a**,**b**) (-)-epicatechin, (**c**,**d**) proanthocyanidin B2, and (**e**,**f**) EGCG monitored within 4 Å space around the ligand in the active pocket of MMP-1. In 2D interaction map, pink arrows represent the hydrogen bonds while green, blue, red, violet, and grey color residues stand for the hydrophobic, polar, negative, positive, and glycine interactions, respectively, in the respective docked complex.

**Figure 6 biomolecules-10-01379-f006:**
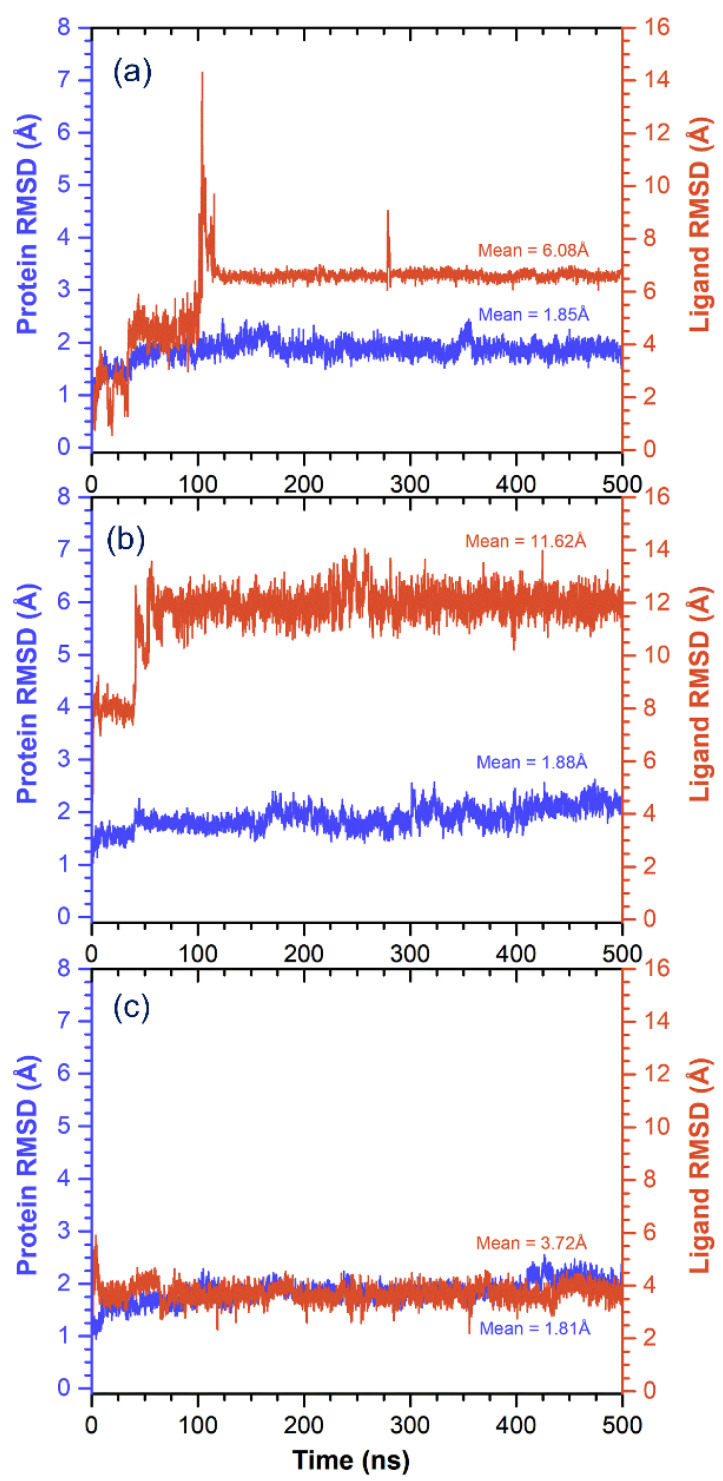
Root mean square deviation (RMSD) values extracted from 500 ns MD simulation trajectories for the docked complexes of MMP-1 with bioactive compounds, viz. (**a**) (-)-epicatechin, (**b**) proanthocyanidin B2, and (**c**) reference compound EGCG.

**Figure 7 biomolecules-10-01379-f007:**
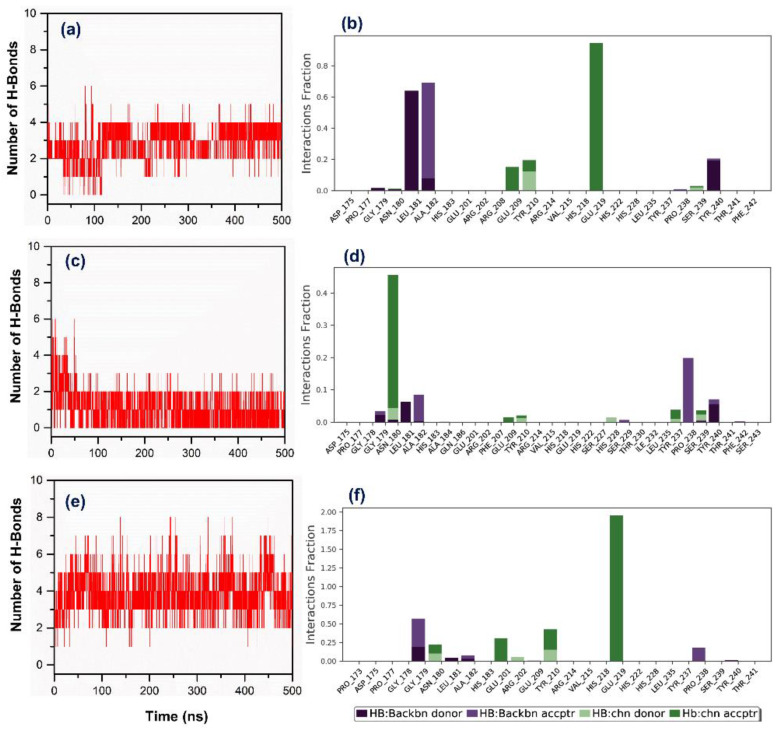
Plots exhibits the total number and specific type of hydrogen bond formation between docked ligands, i.e., (**a**,**b**) (-)-epicatechin, (**c**,**d**) proanthocyanidin B2, and (**e**,**f**) EGCG with MMP-1 during a 500 ns MD simulation interval.

**Figure 8 biomolecules-10-01379-f008:**
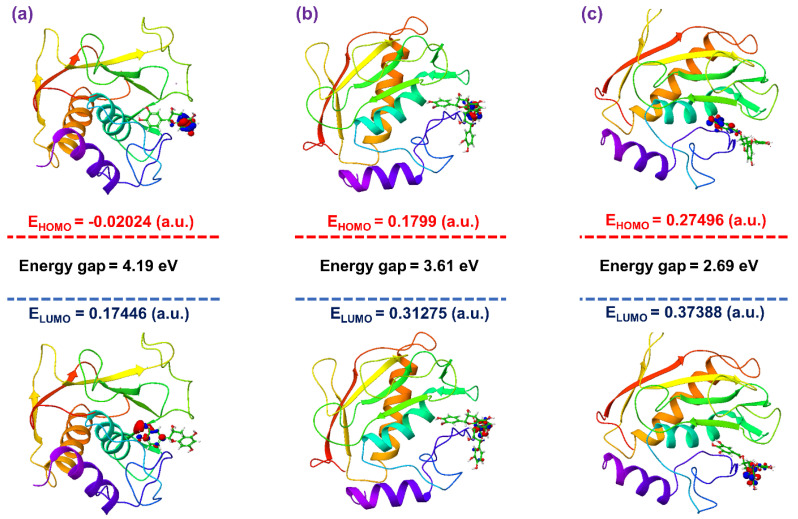
Frontier molecular orbitals, i.e., highest occupied molecular orbitals (HOMO) and lowest unoccupied molecular orbitals (LUMO), along with energy values and energy band gap, were calculated for the molecular docked bioactive compounds (**a**) (-)-epicatechin, (**b**) proanthocyanidin B2, and (**c**) EGCG in the active pocket of MMP-1 using the ONIOM (B3LYP/6-31G**:UFF) method.

**Figure 9 biomolecules-10-01379-f009:**
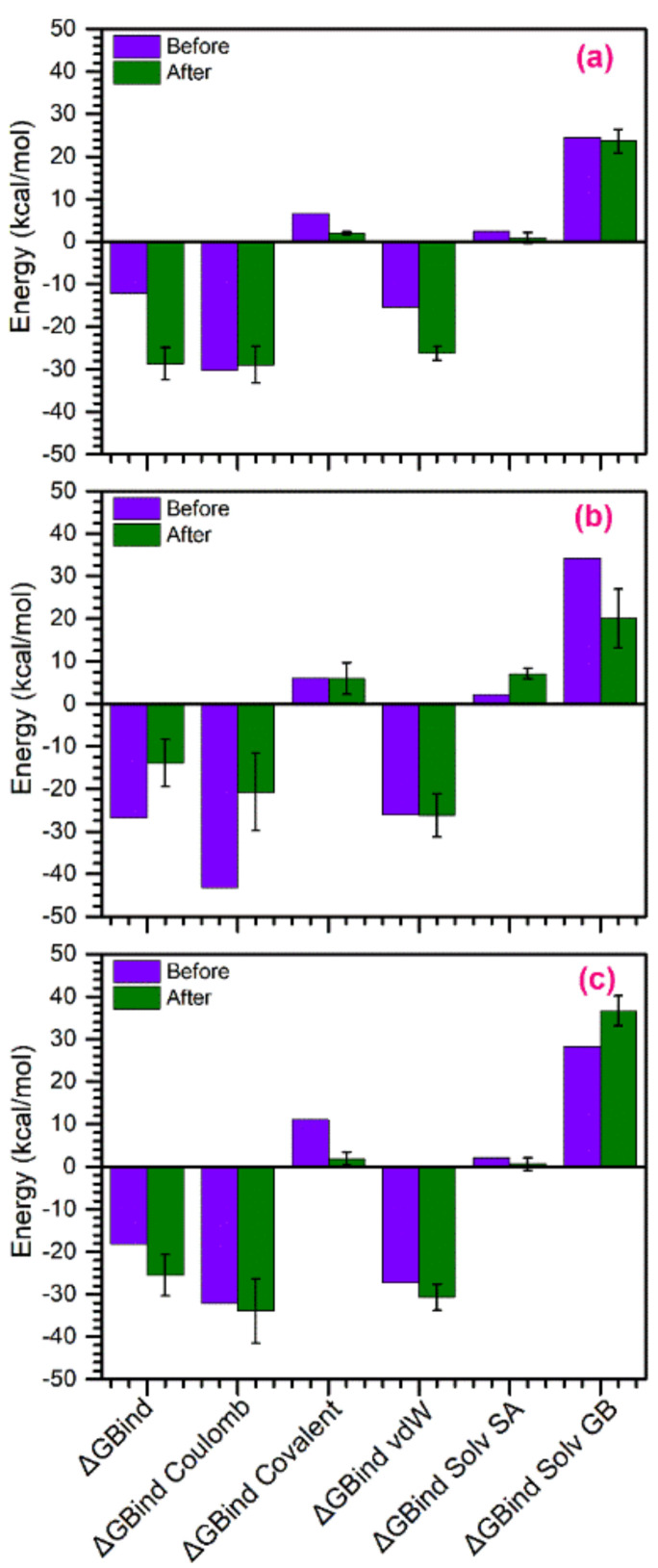
Total Molecular Mechanics Generalized Born Surface Area (MM/GBSA) binding free energy (kcal/mol) values calculated for (**a**) MMP-1–(-)-epicatechin, (**b**) MMP-1–proanthocyanidin B2, and (**c**) MMP-1–EGCG complexes before and after MD simulation.

**Figure 10 biomolecules-10-01379-f010:**
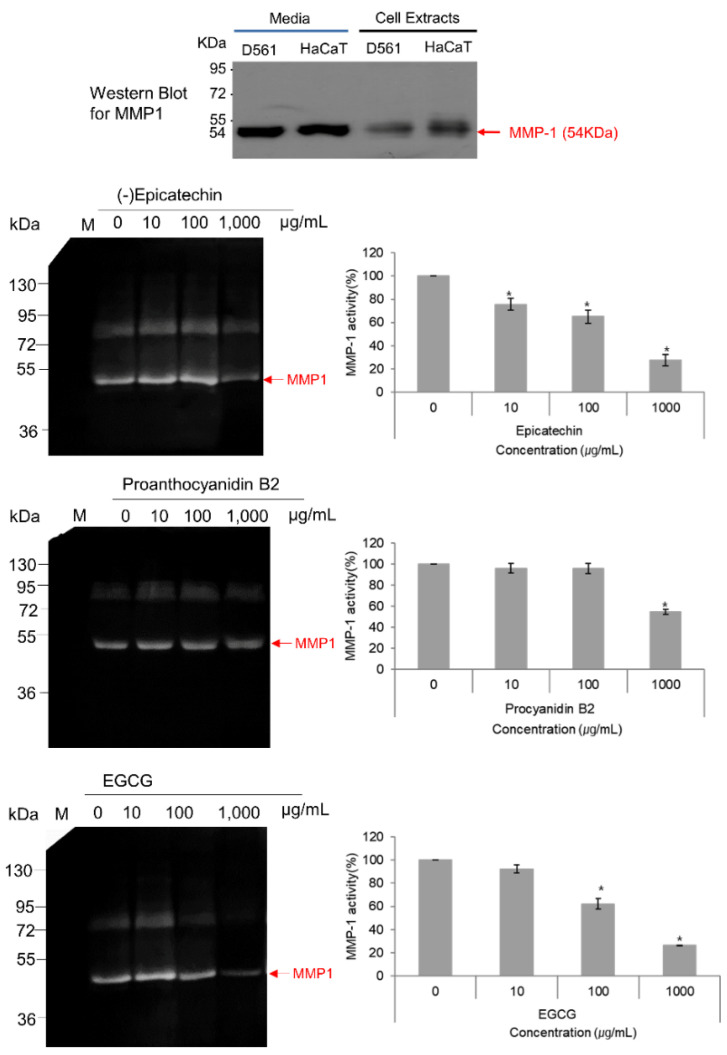
MMP-1 inhibition activity calculated for the selected bioactive compounds, i.e., (-)-epicatechin, proanthocyanidin B2, and EGCG, against beta casein as the substrate: the bar graphs were calculated using LabWorks software based on the width of the inhibition zones in the respective zymographs of triplicate experimental study. Media contained the MMP-1 protein secreted by cells and the human mature MMP-1 protein appeared with antihuman MMP-1-specific antibodies at 54 KDa, as shown in western bolt for MMP-1.

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
