# Peer review of "Computational and In Vitro Investigation of (-)-Epicatechin and Proanthocyanidin B2 as Inhibitors of Human Matrix Metalloproteinase 1"

_biomolecules, 2020, doi:10.3390/biom10101379_

Round 1
Reviewer 1 Report
In this work, the authors used molecular docking and molecular dynamics simulations studies to compare the binding activity of bioactive compounds, epicatechin, proanthocyanidin B2 and epigallocatechin gallate, (EGCG) with MMP-1. EGCG is a well-known inhibitor for MMP-1 and shows a highly active antiaging property. In this work, the in silico results suggested that epicatechin could be a potent inhibitor and antiaging agent like EGCG. The authors have compared the binding properties of epicatechin, proanthocyanidin B2, and (EGCG) at the MMP-1 by comparing hydrogen bonding patterns, van der Waals, polar interactions, metal binding, coulombic, freezing rotatable bonds, and hydrophobic contacts. While I agree that molecular dynamics can provide adequate insight into all of these aspects of small molecule binding to biomacromolecules, the data in this manuscript is too preliminary to support the conclusions or even to paint a robust picture of what is happening.
Although results showed are interesting on the role of bioactive antiaging compounds against MMP-1, still, the current study can be published after the authors address the following important issues with further supporting data and comparison to experimental results.
- At many instances, in the manuscript, the authors raise points about the behavior of small molecules binders but do not provide the data to back up the assertions. For example, the “strength” of a hydrogen bond is mentioned, how is strong vs. weak being defined? How occupied is the specific hydrogen bond? Additional analysis needs to be performed and reported. How often is this particular hydrogen bond occupied in the trajectory? The authors mention hydrogen bonding patterns but do not provide quantitative measurements that are analyzable in simulations.
- While non-covalent binding of small molecules to collagenous is associated with the critical balance of entropy and enthalpy contributions that eventually define the ligand's binding affinity to biomacromolecules. The authors may also correlate the effect of electrostatic, van der Waals interactions of binding of the ligand in the binding pocket with thermodynamic parameters (enthalpic and entropic contributions) and compare the results with experimental values if possible.
Author Response
Response to the Reviewer Comments
#Reviewer 1:
In this work, the authors used molecular docking and molecular dynamics simulations studies to compare the binding activity of bioactive compounds, epicatechin, proanthocyanidin B2 and epigallocatechin gallate, (EGCG) with MMP-1. EGCG is a well-known inhibitor for MMP-1 and shows a highly active antiaging property. In this work, the in-silico results suggested that epicatechin could be a potent inhibitor and antiaging agent like EGCG. The authors have compared the binding properties of epicatechin, proanthocyanidin B2, and (EGCG) at the MMP-1 by comparing hydrogen bonding patterns, van der Waals, polar interactions, metal binding, coulombic, freezing rotatable bonds, and hydrophobic contacts. While I agree that molecular dynamics can provide adequate insight into all of these aspects of small molecule binding to biomacromolecules, the data in this manuscript is too preliminary to support the conclusions or even to paint a robust picture of what is happening.Although results showed are interesting on the role of bioactive antiaging compounds against MMP-1, still, the current study can be published after the authors address the following important issues with further supporting data and comparison to experimental results.
#Comment 1: At many instances, in the manuscript, the authors raise points about the behavior of small molecules binders but do not provide the data to back up the assertions. For example, the “strength” of a hydrogen bond is mentioned, how is strong vs. weak being defined?
Ans. Thank you for your valuable suggestions. In the present study, distance and nature of atoms participating in the hydrogen bond formation were set as criteria to define the strength of hydrogen bond. A suitable discussion and parameters selected to differentiate the strength of H-bond in the docked complexes have been amended in the revised manuscript for your consideration on Page No.16-17, in the second paragraph of Section 3.2. Molecular docking simulation and intermolecular interaction analysis.
#Comment 2: How occupied is the specific hydrogen bond? Additional analysis needs to be performed and reported.
Ans. Thank you for your valuable suggestion. The specificity of the hydrogen bond in the respective docked complexes has been amended in the revised manuscript for your consideration on Page no. 16-18, Section 3.2. Molecular docking simulation and intermolecular interaction analysis.
#Comment 3: How often is this particular hydrogen bond occupied in the trajectory? The authors mention hydrogen bonding patterns but do not provide quantitative measurements that are analyzable in simulations.
Ans. Authors are thankful for your suggestion. We have extracted the particular hydrogen bond occupied in the trajectory along with quantitative measurements, number of hydrogen bond formation per frame during the MD simulation, from the respective simulation trajectories, and has been added in the revised manuscript as Fig. 7 on Page no. 25 under section 3.3. Molecular dynamics simulation analysis.
#Comment 4: While non-covalent binding of small molecules to collagenous is associated with the critical balance of entropy and enthalpy contributions that eventually define the ligand's binding affinity to biomacromolecules. The authors may also correlate the effect of electrostatic, van der Waals interactions of binding of the ligand in the binding pocket with thermodynamic parameters (enthalpic and entropic contributions) and compare the results with experimental values if possible.
Ans. Authors understand the contribution of enthalpy and entropy in binding affinity, however, as suggested in the previous literature and available computational power, we can not conduct the entropy calculations for considered protein-ligand complexes. However, a suitable discussion of entropy and enthalpy on the binding affinities have been incorporated in the methodology section on Page no. 8 under section 2.6 Molecular mechanics generalized born surface area (MM/GBSA) calculations and in results and discussion under section 3.4. Binding free energy calculations on Page no 26-27 for your kind consideration.

Reviewer 2 Report
The manuscript "Computational and in vitro investigation of epicatechin and proanthocyanidin B2 as inhibitor of human matrix metalloproteinase 1," by Lee et al, describes computational and experimental analysis of the activity of some natural products towards the protein MMP-1. The study should be of interest to the drug design community, but I do suggest some changes before publication.
- The reported analysis of the ADME and orbital properties of the ligands optimized in isolation should be compared to an analysis done with the structures the ligands adopt in the binding site. The changes in ligand geometry upon binding can lead to differences in orbitals, orbital energies, and polarity, which can in turn affect their ADME properties.
- both Eq 1 and 2 are generalized equations which bear little relevance to the work. Appropriate equations should be used or they should be removed altogether.
- The function of the "wizards" used in the methods section should be explained.
- The introduction should be either shortened or revised to be more directly relevant to the subject.
Author Response
Response to the Reviewer Comments
#Reviewer 2:The manuscript "Computational and in vitro investigation of epicatechin and proanthocyanidin B2 as inhibitor of human matrix metalloproteinase 1," by Lee et al, describes computational and experimental analysis of the activity of some natural products towards the protein MMP-1. The study should be of interest to the drug design community, but I do suggest some changes before publication.
#Comment 1:The reported analysis of the ADME and orbital properties of the ligands optimized in isolation should be compared to an analysis done with the structures the ligands adopt in the binding site. The changes in ligand geometry upon binding can lead to differences in orbitals, orbital energies, and polarity, which can in turn affect their ADME properties.
Ans. Thank you for giving comments. Authors have followed your given suggestion and change in ligand geometry; orbitals energies and polarity of the docked ligands have been performed in the revised manuscript. Please consider the revision on Page no. 20 for ligand geometry text and FigS1 in the supplementary section, and for orbital energies of ligand in protein environment, please consider the revision on Page no. 7 in methodology section 2.5.Post molecular simulation quantum chemical calculations and under results and discussion on Page no 25-26 with respect to Fig. 8
#Comment 2:Both Eq 1 and 2 are generalized equations which bear little relevance to the work. Appropriate equations should be used, or they should be removed altogether.
Ans. Thank you for bringing in your notice. The Eq 1 has been removed in the revised manuscript while Eq 2 has been suitably modified and discussed under section 2.6. Molecular mechanics generalized born surface area (MM/GBSA) calculations on Page no. 7-8 for your kind consideration.
#Comment 3: The function of the "wizards" used in the methods section should be explained.
Ans. Thank you for your valuable suggestion, authors have included the function with discussion for the wizards in the revised manuscript on Page 5-6 under section 2.3. Molecular docking simulation for your kind consideration.
#Comment 4:The introduction should be either shortened or revised to be more directly relevant to the subject.
Ans. Thank you for your suggestion. We have followed your comment and introduction section has been shortened and revised in the revised manuscript on Page no. 3-4 for your kind consideration.
Reviewer 3 Report
This is a well written report of a thorough study. The methods are up to date, results are presented in a logical systematic fashion and the interpretations are clearly support the findings.
Just one comment:
The authors mentioned a significant role of VdW forces in studied complexes.
In ab initio section an older version of Gaussian program is used, which does not allow to add a missing dispersion term in DFT calculations. An influence of lack of dispersion on geometry and other properties should be discussed in the text.
Author Response
Response to the Reviewer Comments
#Reviewer 3:
This is a well written report of a thorough study. The methods are up to date, results are presented in a logical systematic fashion and the interpretations are clearly support the findings.
Just one comment:
#Comment 1: The authors mentioned a significant role of VdW forces in studied complexes. In ab initio section an older version of Gaussian program is used, which does not allow to add a missing dispersion term in DFT calculations. An influence of lack of dispersion on geometry and other properties should be discussed in the text.
Ans: Thank you for your valuable suggestion. In this study, ab initio DFT calculations have been performed to optimize the ligands and then study their electronic properties with an aim to study the chemical reactivity and for molecular docking analysis. We agreed that van der Waals forces play important roles in the binding of ligands within active site of the protein, therefore, such calculations, including intramolecular, electrostatic and van der Walls(dispersion) energy terms, were conducted via free binding free energy calculations using MM/GBSA method. The methodology in Section 2.6, Pg 6-7 provide the detailed description, and the corresponding results have been discussed in the Section 3.4, Pg 26-27. These calculations provide the required results, which do not necessitate the dispersion terms to be calculated in ab initio calculations. The dispersion corrections have a small effect on optimized geometries but is important in modeling reactions catalyzed by enzymes (J. Chem. Theory Comput. 2012, 8, 4637-4645); this reference is also discussed in the main text for your kind consideration.
Round 2
Reviewer 1 Report
The authors have revised the manuscript according to my initial suggestions and have addressed the questions in my initial review. I have no further issues with the manuscript.